# Erythema Increase Predicts Psoriasis Improvement after Phototherapy

**DOI:** 10.3390/jcm10173897

**Published:** 2021-08-30

**Authors:** Trinidad Montero-Vilchez, Antonio Martinez-Lopez, Alvaro Sierra-Sanchez, Miguel Soler-Gongora, Eladio Jimenez-Mejias, Alejandro Molina-Leyva, Agustin Buendia-Eisman, Salvador Arias-Santiago

**Affiliations:** 1Dermatology Department, Hospital Universitario Virgen de las Nieves, Avenida de Madrid, 15, 18012 Granada, Spain; tmonterov@correo.ugr.es (T.M.-V.); alejandromolinaleyva@gmail.com (A.M.-L.); salvadorarias@ugr.es (S.A.-S.); 2Instituto de Investigación Biosanitaria GRANADA, 18012 Granada, Spain; alvarosisan@gmail.com; 3Dermatology Department, Faculty of Medicine, University of Granada, 18001 Granada, Spain; miguelsg@correo.ugr.es (M.S.-G.); abuendia@ugr.es (A.B.-E.); 4Epidemiology and Public Health Department, Faculty of Medicine, University of Granada, 18012 Granada, Spain; eladiojimenez@ugr.es

**Keywords:** phototherapy, psoriasis, skin barrier, skin physiology, skin homeostasis

## Abstract

Psoriasis is a major global health problem. There is a need to develop techniques to help physicians select the most appropriate cost-effective therapy for each patient. The main objectives of this study are (1) to evaluate changes in epidermal barrier function and skin homeostasis after phototherapy and (2) to explore potentially predictive values in epidermal barrier function and skin homeostasis to assess clinical improvement after fifteen sessions of phototherapy. A total of 76 subjects, 38 patients with plaque-type psoriasis and 38 gender- and age-matched healthy volunteers, were included in the study. Erythema, transepidermal water loss (TEWL), temperature, stratum corneum hydration (SCH), pH, sebum, and antioxidant capacity were measured before and after the first and fifteenth phototherapy session. Erythema (401.09 vs. 291.12 vs. 284.52 AU, *p* < 0.001) and TEWL (18.23 vs. 11.44 vs. 11.41 g·m^−2^·h^−1^, *p* < 0.001) were significantly higher at psoriatic plaques than in uninvolved psoriatic skin and healthy volunteers, respectively, while SCH was lower (9.71 vs. 44.64 vs. 40.00 AU, *p* < 0.001). After fifteen phototherapy sessions, TEWL (–5.19 g·m^−2^·h^−1^, *p* = 0.016) decreased while SCH (+7.01 AU, *p* = 0.013) and erythema (+30.82 AU, *p* = 0.083) increased at psoriatic plaques. An erythema increase exceeding 53.23 AU after the first phototherapy session, with a sensitivity of 71.4% and specificity of 84.2%, indicates that a patient may improve Psoriasis Area and Severity Index (PASI) by ≥3 points after fifteen phototherapy sessions. In conclusion, phototherapy improves epidermal barrier function in psoriatic patients and the erythema increase after one phototherapy session could help doctors select psoriasis patients who are more likely to respond to phototherapy.

## 1. Introduction

Psoriasis is a chronic, recurrent, multisystemic inflammatory disease [1] caused by a combination of immunological imbalances, genetic associations, and environmental factors [2]. Its prevalence around the word has been estimated at between 0.51% and 11.43% [3]. Psoriasis is considered a major global health problem [4]. Although the skin manifestations are often the only recognized symptoms of psoriasis [5], this disease is associated with multiple comorbidities [6,7,8,9] and impacts the patient’s quality of life [5,10]. Moreover, the economic burden of psoriasis is high, as in Europe the annual total cost per patient is EUR 6000–12,000 [11].

Multiple treatments are effective for psoriasis, including topical medicines, oral systemic prescriptions, phototherapy, and biologics [12]. Nevertheless, it is not known which type of patient would respond best to each treatment [13]. Moreover, tools to assess disease severity and treatment effectiveness are subjective [14]. Thus, there is a need to develop techniques to help physicians select the most appropriate cost-effective therapy for each patient [15].

Phototherapy is an effective, safe, and low-cost therapy for mild–moderate psoriasis, although many medical appointments are needed to see an improvement [16]. Several types of light and lasers have been developed to treat psoriasis, the narrowband ultraviolet light B (NB-UVB) being the most frequently used. NB-UVB wavelengths ranges from 311 to 313 nm. The starting dose is based on skin phototype or minimal erythema dose (MED), and two or three sessions per week are recommended [17]. Selecting the right patient profile for this treatment and accurately assessing disease severity would improve patient satisfaction and healthcare spending [13]. It would also be interesting to predict the response to assess home phototherapy effectiveness [18]. As the development of psoriasis plaques results from the deregulation of epidermal keratinocytes and immunity cells [19] and the phototherapy’s beneficial effect on psoriasis lesions is explained by it blocking epidermal hyperproliferation and an immunomodulatory effect [20], objective changes in the epidermal barrier function may help to select the right psoriasis patients for phototherapy treatment and to assess disease improvement. Epidermal barrier dysfunction in psoriasis patients has previously been reported, assessed by an increase in transepidermal water loss (TEWL) and a decrease in stratum corneum hydration (SCH) [21,22]. To date, only three studies have evaluated the variations in epidermal barrier function following phototherapy, displaying an improvement in TEWL and SCH [23,24,25].

Thus, the aims of this study are (1) to compare epidermal barrier function and skin homeostasis of healthy volunteers, uninvolved psoriatic skin, and psoriatic plaques, (2) to assess changes in epidermal barrier function and skin homeostasis after one session of phototherapy, (3) to assess changes in epidermal barrier function and skin homeostasis after fifteen phototherapy sessions, and (4) to explore potentially predictive values in epidermal barrier function and skin homeostasis to assess clinical improvement after fifteen phototherapy sessions.

## 2. Materials and Methods

### 2.1. Design

A cross-sectional study was conducted to evaluate epidermal barrier function and skin homeostasis disparities between healthy skin, uninvolved psoriatic skin, and psoriatic plaques.

A prospective observational study was carried out on patients with psoriasis to assess epidermal barrier function and skin homeostasis following fifteen phototherapy sessions. Psoriatic patients were exposed to fifteen phototherapy sessions, while healthy volunteers were only reviewed after this period of time without being exposed to phototherapy.

### 2.2. Setting

This study was conducted between September 2019 and March 2020 in the Dermatology Department of the Hospital Universitario Virgen de las Nieves in Granada, Spain.

### 2.3. Study Population

Inclusion Criteria:Patients with established clinical diagnosis of active moderate-to-severe plaque-type psoriasis (minimum Psoriasis Area and Severity Index (PASI) score of 4) [1] selected by clinical criteria to attend phototherapy treatment with UVB narrowband (NB-UVB) [16].Controls were healthy volunteers, gender- and age-matched (±3 years) with psoriasis patients. These volunteers were people who attended the Dermatology Department for trivial conditions such as melanocytic nevi or seborrheic keratoses. The same criteria were used to select the non-exposed group in the prospective study.

Exclusion Criteria:For psoriasis patients, currently having non-plaque forms of psoriasis.For healthy volunteers, having previous personal or family history of any inflammatory skin disease.Clinical infection on the treatment area.History of cancer or an immunocompromised disease.Not signing the informed consent form.

### 2.4. Follow-Up and Exposure

Exposed subjects were evaluated before and after receiving the first phototherapy session and before and after the 15th phototherapy session. The starting dose for NB-UVB therapy and the dosage schedule were based on skin phototype following the current guidelines [17]. The frequency was two or three times a week depending on the patient’s availability. Non-exposed subjects were evaluated twice, on the same days as their exposed pair.

### 2.5. Variables

Clinical and sociodemographic variables. Gender, age, smoking and alcohol habit, psoriasis family history, and use of emollients were gathered by means of clinical interview. Psoriasis severity was assessed by the PASI and the body surface area (BSA). Every study patient was also evaluated with the Dermatology Life Quality Index (DLQI). Information about disease duration, previous treatment, the previous number of phototherapy sessions, session dose, and total cumulative dose was also collected.Epidermal barrier function variables. Homeostasis parameters related to epidermal barrier function and skin homeostasis were measured. SCH (in arbitrary units, using Corneometer^®^CM825), TEWL (in g·m^−2^·h^−1^, using Tewameter^®^TM300), pH (using Skin-pH-Meter^®^PH905), erythema index (in arbitrary units, using Mexameter^®^MX18), sebum (in arbitrary units, usingSebumeter^®^SM815), and skin temperature (in °C, using Skin-ThermometerST500) were measured by a Multi Probe Adapter (MPA, Courage + Khazaka electronic GmbH, Köln, Germany). Total antioxidant capacity (TAC) was measured using eBQC^®^ electrochemical method (Bioquochem S.L. (BQCkit), Asturias, Spain), and expressed in microcoulombs. TAC is divided into two sections: fast antioxidants (Q1), which have a lower oxidation potential, and slow antioxidants (Q2) [26]. All variables were measured at a psoriatic plaque on the elbow and at an uninvolved skin area near the elbow in psoriatic patients, while healthy subjects were measured at a skin area on their elbows. All parameters were measured ten times for each area, using their average for analysis. The measurements were taken in the same room. The average ambient air temperature at the time of the study was 22 ± 1°C, and the average ambient air humidity was 45% ± 3%.

### 2.6. Statistical Analysis

Descriptive statistics were used to present the sample characteristics. Continuous data were expressed as the mean ± standard deviation. The absolute and relative frequency distributions were estimated for qualitative variables. The Shapiro–Wilk test was used to check the normality of data distribution, and Levene’s test was used to check the homogeneity of variance. Linear regression models were constructed to compare continuous data between healthy skin and psoriatic patients. To predict PASI improvement after fifteen phototherapy sessions, cut-off points were generated using ROC curves for the changes of erythema and SCH after the first phototherapy session. To produce these ROC curves, the sensitivities and specificities for changes of erythema and SCH values after the first phototherapy that predict an improvement in PASI of ≥3 after the fifteenth phototherapy session were tabulated and the graphical ROC curve was generated by plotting true positive rate (sensitivity) on the *y*-axis against false positive rate (1-specificity) on the *x*-axis for the various values tabulated. To select the optimal cut-off point, the point nearest to the top-left-most corner of the ROC curve was chosen, giving equal weight to the importance of sensitivity and specificity. A *p*-value of <0.05 was considered statistically significant. Statistical Analyses were performed using the SPSS package (SPSS for Windows, Version 24.0 Chicago, IL, USA: SPSS Inc.).

### 2.7. Ethics

This study was authorized by the ethics committee of Hospital Universitario Virgen de las Nieves. The nature of the study was explained to all participants, who agreed to participate through verbal and written consent. The measurements taken were noninvasive, and patient data were kept confidential. All experiments were done in accordance with relevant guidelines and regulations.

## 3. Results

### 3.1. Skin Homeostasis Parameters between Healthy Participants and Psoriatic Patients

The study included 76 participants, consisting of 38 psoriatic patients and 38 healthy participants, Appendix A.

Differences in skin homeostasis parameters between healthy skin, uninvolved, and involved psoriatic skin before phototherapy were found, Table 1. Lower TEWL values were found in healthy skin compared with uninvolved psoriatic skin and psoriatic plaques (11.41 vs. 11.44 vs. 18.23 g·m^−2^·h^−1^, *p* < 0.001). Higher SCH values were observed in healthy skin compared with uninvolved psoriatic skin and psoriatic plaques (40.00 vs. 44.64 vs. 9.71 AU, *p* < 0.001). Lower temperature values were detected in uninvolved psoriatic skin than at psoriatic plaques (30.40 vs. 31.25 °C, *p* < 0.001). Lower erythema index was found in healthy skin than in uninvolved psoriatic skin and psoriatic plaques (284.52 vs. 291.12 vs. 401.09 AU, *p* < 0.001). Higher total antioxidant capacity was observed in uninvolved psoriatic skin than at psoriatic plaques (6.33 vs. 5.54 uC, *p* = 0.014). No differences were found in pH or sebum.

### 3.2. Differences in Skin Homeostasis Parameters after One Phototherapy Session

Skin homeostasis parameters were modified after one phototherapy session, Table 2. TEWL did not change at psoriatic plaques or in uninvolved skin after one phototherapy session. The effect of phototherapy on SCH values was different depending on the skin involvement. It was observed that SCH increased by 2.45 ± 0.72 AU (*p* = 0.002) at psoriatic plaques (but SCH was not modified in uninvolved skin (*p* = 0.126).

Temperature increased by 0.24 ± 0.10 °C at psoriatic plaques (*p* = 0.016). The erythema index increased by 31.42 ± 8.30 AU (*p* < 0.001) at psoriatic plaques, but no changes were observed in uninvolved skin.

Total antioxidant capacity was not modified at psoriatic plaques or in uninvolved skin after one phototherapy session. No differences in pH or sebum were observed.

### 3.3. Skin Homeostasis Changes after Follow-Up

The prospective study included 76 subjects, where 52 (68.42%) met the requirements (26 psoriatic patients and 26 healthy participants). The mean session dose at baseline was 0.46 (0.31) J. Homeostasis parameters changed after follow-up, Table 3. TEWL decreased by 3.50 ± 1.41 g·m^−2^·h^−1^ in uninvolved skin (*p* = 0.021) and by 5.19 ± 2.00 g·m^−2^·h^−1^ at psoriatic plaques (*p* = 0.016). No effect was observed in healthy non-exposed skin. SCH increased by 7.01 ± 2.63 AU at psoriatic plaque (*p* = 0.013), while no changes were observed in healthy skin.

Temperature increased after phototherapy by 1.5 ± 0.26 °C in uninvolved skin (*p* < 0.001) and by 1.42 ± 0.28 °C at psoriatic plaques (*p* < 0.001), while it did not change in healthy non-exposed skin. Erythema increased by 31.83 ± 17.06 AU in uninvolved skin (*p* = 0.007), and an almost significant increase of 30.82 ± 17.06 AU was also observed at psoriatic plaques (*p* = 0.087), Figure 1.

### 3.4. Skin Homeostasis Predicts PASI Improvement

After follow-up, PASI decreased by 3.13 ± 3.13 points, so patients were placed in two groups: PASI reduction < 3 and PASI reduction ≥ 3. Of the patients, 73.1% (19/26) were included in the first group and 26.9% (7/26) in the second. After the first phototherapy session, patients with a PASI improvement ≥ 3 showed a higher erythema increase (71.08 vs. 11.54 AU, *p* = 0.011), and an almost significant higher SCH increase (4.69 vs. 1.40; *p* = 0.141) and higher TEWL decrease (−4.97 vs. 0.86 g·m^−2^·h^−1^, *p* = 0.199).

A ROC curve was generated to determine an optimum cut-off value for erythema increases after one phototherapy session, which allowed clinical improvement after 15 phototherapy sessions to be predicted (area under the curve = 0.789, *p* = 0.026) (Figure 2A). A value for erythema increases exceeding 53.23 AU after the first phototherapy session, with a sensitivity of 71.4% and specificity of 84.2%, indicates that a patient may improve PASI by ≥3 points after fifteen phototherapy sessions.

SCH increases were also higher in patients with PASI improvement ≥ 3. An ROC curve was generated to determine an optimum cut-off value for SCH increase after one phototherapy session, which allowed clinical improvement after 15 phototherapy sessions to be predicted (area under the curve = 0.692, *p* = 0.1402) (Figure 2B). A value for SCH increases exceeding 1.06 AU after the first phototherapy session, with a sensitivity of 71.4% and specificity of 63.8%, indicates that a patient may improve PASI by ≥3 points after fifteen phototherapy sessions.

After calculating the different cut-off levels, we evaluated whether combined values may also predict clinical improvement. Patients with erythema increase > 53.23 AU and SCH increase > 1.06 AU after the first phototherapy session may improve PASI by ≥3 after 15 phototherapy sessions, with a sensitivity of 57.1% and a specificity of 94.7% (Appendix A).

## 4. Discussion

Differences in skin homeostasis parameters between healthy skin, uninvolved psoriatic skin, and psoriatic plaques have been observed. After one phototherapy session, temperature, erythema, and SCH increased at psoriatic plaques. Moreover, after fifteen phototherapy sessions, decreased TEWL and increased SCH and temperature levels at psoriatic plaques were observed. Phototherapy could improve epidermal barrier function and skin homeostasis in psoriatic patients, and erythema increases after one phototherapy session could help clinicians select psoriasis patients with more probability of responding to phototherapy.

In agreement with previous reports, it has been observed that the whole epidermal barrier is affected in psoriatic patients, not only at psoriatic plaques [27]. Other research also found higher TEWL at psoriatic plaques than in uninvolved psoriatic skin and healthy controls [21,22,27] and lower SCH values at psoriatic plaques than in uninvolved psoriatic skin and healthy controls [21,27,28]. The differences in TEWL and SCH values between psoriatic plaques and uninvolved psoriatic skin may be explained by a low AQP3 expression in plaques [29]. Temperature and erythema were also higher at psoriatic skin, probably due to its inflammatory pathogenesis [30]. Moreover, TEWL and temperature at psoriatic plaques were noted as useful tools for evaluating psoriasis severity [27].

The role of phototherapy on epidermal barrier function and skin homeostasis is not well known. Our results found an improvement in epidermal barrier function and skin homeostasis after phototherapy. Recently, it has been observed that SCH decreased, and TEWL, erythema, and temperature increased at psoriatic plaques after only one phototherapy session [25]. Moreover, it was shown that phototherapy increased SCH and decreased TEWL after fourteen [24] and twenty-four [23] phototherapy sessions, without information regarding other skin homeostasis parameters. Our study found increased SCH at psoriatic plaques following only one phototherapy session and increased SCH and decreased TEWL at psoriatic plaques after fifteen phototherapy sessions. Moreover, in contrast with previous studies, we also included a non-exposed group with follow-up to prove that changes in SCH are not because of time. Changes in SCH might be due to the inhibition of epidermal hyperproliferation caused by phototherapy [20,31]. SCH and TEWL changes were greater at psoriatic plaques than in uninvolved psoriatic skin, which might underline a local effect on psoriasis plaques [32,33]. Temperature and erythema index rose after the phototherapy session, in agreement with previous reports [25,34,35,36]. Assessment of temperature and erythema increase may help clinicians optimize phototherapy to treat patients with an effective dosage without adverse events. The pH increased in healthy skin, uninvolved psoriatic skin, and psoriatic plaques, suggesting that time may have an effect on pH changes. Antioxidant capacity also decreased in healthy skin, uninvolved psoriatic skin, and psoriatic plaque. This fact might mean that the time have also an impact in antioxidant capacity or that the sticks used might lose their capacity to measure the antioxidant capacity along the time. There is little information regarding the effect of phototherapy on antioxidant capacity. Oxidative stress has been evaluated by measuring different parameters of a blood sample, with controversial results. Darlenski et al. found a slight decrease in the detoxifying activity of catalase without significant differences after phototherapy [24]. On the other hand, Pektas et al. observed total oxidant status and oxidative stress index increased after phototherapy [37]. Our results showed total antioxidant capacity decreases after phototherapy, in agreement with this research by Pektas.

Brazzelli et al. suggested that SCH improvement at psoriatic plaques might precede clinical improvement [23]. As far as we know, it is not known which parameters might predict clinical improvement in psoriatic patients treated with phototherapy. We observed that SCH changes after one phototherapy session might predict PASI improvement after fifteen phototherapy sessions. Moreover, a value for erythema increases exceeding 53.23 AU after the first phototherapy session, with a sensitivity of 71.4% and specificity of 84.2%, indicates that a patient may improve PASI by ≥3 points after fifteen phototherapy sessions. This research could help clinicians select psoriatic patients for phototherapy treatment. Therefore, patients who do not reach this value of erythema after the first session can be treated with another therapeutic alternative. Moreover, this research would also be interesting for selecting candidates for home phototherapy, as patients who have an erythema increase exceeding 53.23 AU after the first phototherapy session may improve during treatment.

This study has some limitations. (1) The variation of the homeostasis parameters depending on external conditions. Nevertheless, to improve outcome reliability, all participants were measured by the same researcher in the same room and the ambient conditions were measured. (2) The loss of patients observed during follow-up as COVID-19 broke out during the follow-up period and the activity of dermatology practices was greatly reduced.

## 5. Conclusions

As far as we know, this is the first study to propose a cut-off point in erythema increases after one phototherapy session to select psoriasis patients with more likelihood of responding to fifteen phototherapy sessions. This could increase the treatment’s cost-effectiveness and reduce indirect costs and hospital visits for patients with probable low response.

## Figures and Tables

**Figure 1 jcm-10-03897-f001:**
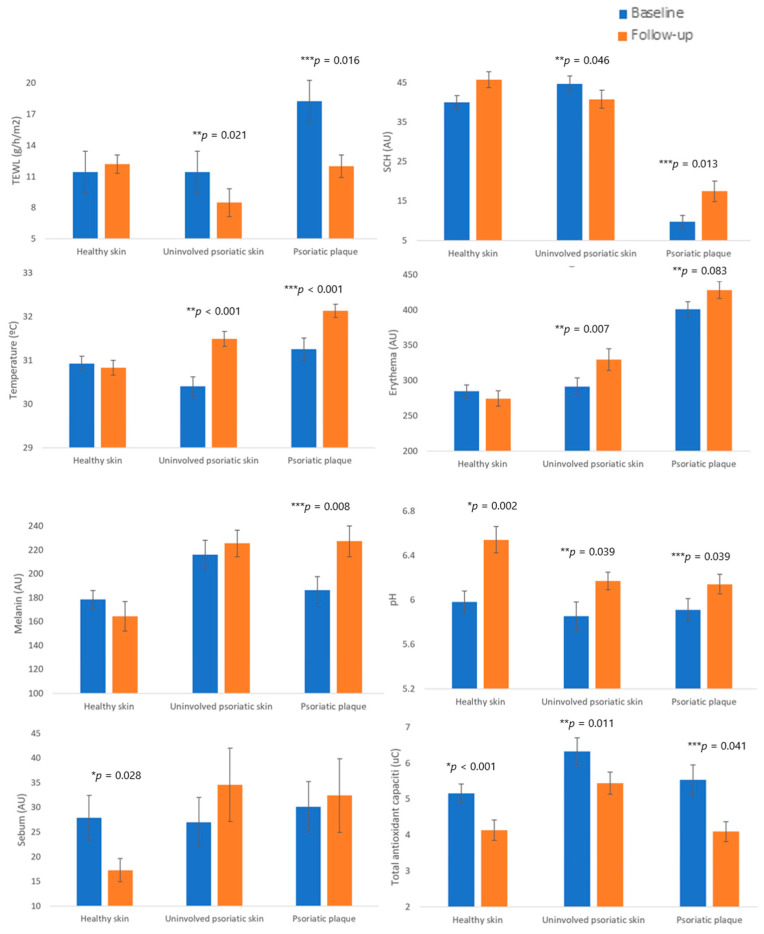
Homeostasis skin parameters in healthy skin, uninvolved psoriatic skin, and psoriatic plaques before and after follow-up. AU, arbitrary units; SCH, stratum corneum hydration; TEWL, transepidermal water loss; uC, microcoulombs. * *p-*value after using a linear regression model adjusted by emollient use to compare homeostasis parameters between control and uninvolved psoriatic skin before phototherapy. ** *p-*value after using a linear regression model adjusted by emollient use to compare homeostasis parameters between control and psoriatic plaques before phototherapy. *** *p-*value after using Student’s *t*-test for paired samples to compare homeostasis parameters between uninvolved psoriatic skin and psoriatic plaques before phototherapy. Only *p*-values of <0.05 are shown.

**Figure 2 jcm-10-03897-f002:**
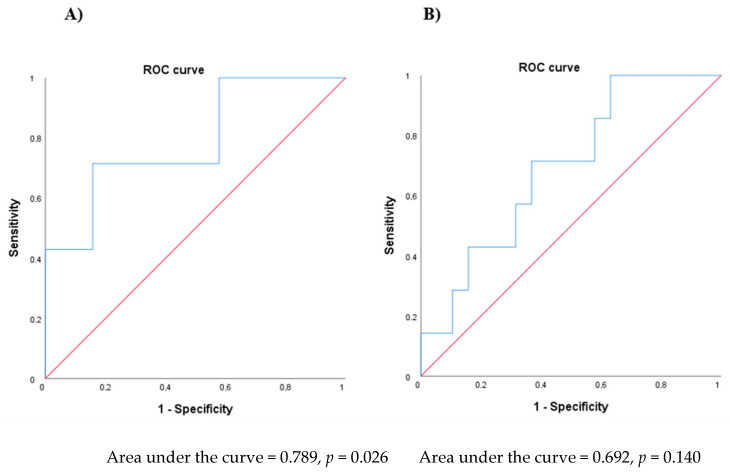
Receiver operating characteristic (ROC) curve for the values of erythema increases after one phototherapy session. (**A**) A receiver operating characteristic (ROC) curve was created to determine the optimal cut-off value of erythema increases after one phototherapy session to predict PASI improvement in patients with psoriasis after fifteen phototherapy sessions (area under curve = 0.789, *p* = 0.026). An erythema increase exceeding 53.23 AU after the first phototherapy session had high probability of improving PASI by ≥3 points after fifteen phototherapy sessions (sensitivity = 71.4%; specificity = 84.2%). (**B**) A receiver operating characteristic (ROC) curve was created to determine the optimal cut-off value of stratum corneum hydration (SCH) increases after one phototherapy session to predict PASI improvement in patients with psoriasis after fifteen phototherapy sessions (area under curve = 0.692, *p* = 0.140). An SCH increase exceeding 1.06 AU after the first phototherapy session had high probability of improving PASI by ≥3 points after fifteen phototherapy sessions (sensitivity = 71.4%; specificity = 63.8%).

**Table 1 jcm-10-03897-t001:** Homeostasis parameters in healthy skin, uninvolved psoriatic skin, and involved psoriatic skin at baseline.

	Healthy Skin at Baseline (*n* = 38)	Uninvolved Psoriatic Skin at Baseline (*n* = 38)	Psoriatic Plaques at Baseline (*n* = 38)	*p* *	*p* **	*p* ***
TEWL (g·m^−2^h^−1^)	11.41 (6.63)	11.44 (8.11)	18.23 (9.46)	0.792	<0.001 **	<0.001 ***
SCH (AU)	40.00 (10.50)	44.64 (12.49)	9.71 (9.81)	0.073	<0.001 **	<0.001 ***
Temperature (°C)	30.92 (1.04)	30.40 (1.34)	31.25 (1.59)	0.080	0.280	<0.001 ***
Erythema (AU)	284.52 (55.54)	291.12 (75.43)	401.09 (64.51)	0.574	<0.001 **	<0.001 ***
pH	5.98 (0.63)	5.86 (0.64)	5.91 (0.47)	0.321	0.301	0.728
Sebum (AU)	27.91 (26.95)	26.97 (30.50)	30.14 (30.38)	0.957	0.056	0.386
Q1 (uC)	0.86 (0.2)	1.15 (0.46)	0.96 (0.45)	0.001 *	0.176	0.001 ***
Q2 (uC)	4.30 (1.37)	5.20 (1.85)	4.57 (2.16)	0.028 *	0.565	0.026 ***
QT (uC)	5.16 (1.53)	6.33 (2.26)	5.54 (2.53)	0.015 *	0.474	0.014 ***

AU, arbitrary units; Q1, fast antioxidant capacity; Q2, slow antioxidant capacity; QT, total antioxidant capacity; SCH, stratum corneum hydration; TEWL, transepidermal water loss; uC, microcoulombs. The data are expressed as means (standard deviation). * *p-*value after using a linear regression model adjusted by emollient use to compare homeostasis parameters between healthy skin and uninvolved psoriatic skin at baseline. ** *p-*value after using a linear regression model adjusted by emollient use to compare homeostasis parameters between healthy skin and psoriatic plaques at baseline. *** *p-*value after using Student’s *t*-test for paired samples to compare homeostasis parameters between uninvolved psoriatic skin and psoriatic plaques at baseline.

**Table 2 jcm-10-03897-t002:** Homeostasis parameters for uninvolved psoriatic skin and psoriatic plaques after one phototherapy session.

	Uninvolved Psoriatic Skin after One Phototherapy Session (*n* = 38)	Psoriatic Plaques after One Phototherapy Session (*n* = 38)	Mean Difference in Uninvolved Skin after vs. before Phototherapy	Mean Difference at Psoriatic Plaques after vs. before Phototherapy	*p* *	*p* **	*p* ***
TEWL (g·m^−2^h^−1^)	10.78 (8.84)	17.72 (8.46)	−0.66 (0.87)	−0.52 (0.94)	<0.001 *	0.45	0.568
SCH (AU)	42.78 (11.26)	12.16 (10.77)	−1.86 (1.19)	2.45 (0.72)	<0.001 *	0.126	0.002 ***
Temperature (°C)	30.54 (1.54)	31.49 (1.42)	0.14 (0.13)	0.24 (0.1)	<0.001 *	0.297	0.016 ***
Erythema (AU)	294.11 (78.14)	432.51 (81.91)	2.98 (6.19)	31.42 (8.30)	<0.001 *	0.633	0.001 ***
pH	5.84 (0.54)	6.04 (0.51)	−0.03 (0.11)	0.13 (0.18)	0.081	0.815	0.1
Sebum (AU)	30.21 (27.40)	27.71 (17.19)	0.97 (3.65)	−3.41 (3.84)	0.571	0.792	0.381
Q1 (uC)	1.09 (0.32)	0.93 (0.39)	−0.05 (0.07)	−0.03 (0.06)	0.010 *	0.42	0.66
Q2 (uC)	5.00 (1.31)	4.51 (1.57)	−0.18 (0.26)	−0.09 (0.24)	0.026 *	0.494	0.744
QT (uC)	6.09 (1.55)	5.44 (1.91)	−0.21 (0.32)	−0.11 (0.28)	0.013 *	0.505	0.703

AU, arbitrary units; Q1, fast antioxidant capacity; Q2, slow antioxidant capacity; QT, total antioxidant capacity; SCH, stratum corneum hydration; TEWL, transepidermal water loss; uC, microcoulombs. The data is expressed are means (standard deviation). * *p-*value after using Student’s *t*-test for paired samples to compare homeostasis parameters between uninvolved psoriatic skin and psoriatic plaques after one phototherapy session. ** *p-*value after using Student’s *t*-test for paired samples to compare homeostasis parameters in uninvolved psoriatic skin before and after one phototherapy session. *** *p-*value after using Student’s *t*-test for paired samples to compare homeostasis parameters at psoriatic plaques before and after one phototherapy session.

**Table 3 jcm-10-03897-t003:** Homeostasis parameters in healthy skin, uninvolved psoriatic skin, and psoriatic plaques after fifteen phototherapy sessions.

	Healthy Skin after Follow-Up (*n* = 26)	Uninvolved Psoriatic Skin after Phototherapy (*n* = 26)	Psoriatic Plaques after Phototherapy (*n* = 26)	Mean Difference in Healthy Skin after Follow-Up	Mean Difference in Uninvolved Skin after vs. before Phototherapy	Mean Difference at Psoriatic Plaques after vs. before Phototherapy	*p* *	*p* **	*p* ***
TEWL (g·m^−2^h^−1^)	12.18 (4.5)	8.48 (6.77)	11.98 (5.45)	0.30 (1.11)	−3.50 (1.41)	−5.19 (2.00)	0.786	0.021 **	0.016 ***
SCH (AU)	45.73 (10.13)	40.78 (11.70)	17.45 (13.41)	4.18 (1.96)	−6.10 (2.91)	7.01 (2.63)	0.53	0.046 **	0.013 ***
Temperature (°C)	30.93 (1.39)	31.49 (0.88)	32.13 (0.75)	−0.01 (0.25)	1.5 (0.26)	1.42 (0.28)	0.537	<0.00 1 **	<0.001 ***
Erythema (AU)	274.13 (55.65)	329.57 (79.44)	428.15 (61.82)	−13.50 (6.78)	31.83 (17.06)	30.82 (17.06)	0.68	0.007 *	0.083
pH	6.54 (0.59)	6.20 (0.28)	6.26 (0.36)	0.65 (0.18)	0.37 (0.16)	0.37 (0.16)	0.002 *	0.039 **	0.039 ***
Sebum (AU)	17.27 (11.90)	35.00 (39.18)	32.26 (38.78)	−10.38 (4.45)	12.83 (9.51)	7.17 (9.80)	0.028 *	0.190	0.472
Q1 (uC)	0.72 (0.27)	0.87 (0.25)	0.85 (0.30)	−0.17 (0.04)	−0.39 (0.09)	−0.22 (0.11)	<0.001 *	<0.001 **	0.059
Q2 (uC)	3.39 (1.20)	4.32 (1.37)	4.07 (1.20)	−1.03 (0.26)	−1.24 (0.43)	−1.01 (0.49)	0.001 *	0.009 *	0.049 *
QT (uC)	4.13 (1.42)	5.44 (1.57)	4.90 (1.43)	−1.18 (0.27)	−1.36 (0.49)	−1.23 (0.57)	<0.001 *	0.011 *	0.041 *

AU, arbitrary units; Q1, fast antioxidant capacity; Q2, slow antioxidant capacity; QT, total antioxidant capacity; SCH, stratum corneum hydration; TEWL, transepidermal water loss; uC, microcoulombs. The data are expressed as means (standard deviation). * *p-*value after using Student’s *t*-test for paired samples to compare homeostasis parameters in healthy skin before and after the follow-up. ** *p-*value after using Student’s *t*-test for paired samples to compare homeostasis parameters in uninvolved psoriatic skin before and after fifteen phototherapy sessions. *** *p-*value after using Student’s *t*-test for paired samples to compare homeostasis parameters at psoriatic plaques before and after fifteen phototherapy sessions.

## Data Availability

The data presented in this study are available on request from the corresponding author.

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
