# Peer review of "Erythema Increase Predicts Psoriasis Improvement after Phototherapy"

_jcm, 2021, doi:10.3390/jcm10173897_

Round 1

Reviewer 1 Report

Dear Authors,

In the manuscript by Trinidad Montero-Vilchez et al, on “Erythema increase predicts psoriasis improvement after phototherapy: an observational prospective study”, the authors have performed extensive research, but some methodological and outcome questions have risen.

Also, probably due to the language barrier, many sentences and ideas are not clear or easy to understand. Additionally, some sentences are too long. The manuscript needs major revisions. Rephrase:

  1. Line 8
  2. Lines 46-48
  3. Line 60
  4. Lines 69-71
  5. Line 88 UVB-NB or NB-UVB Line 103?
  6. Lines 104-106.
  7. Line 178
  8. Lines 270-272
  9. Lines 279-280
  10. Lines 291-293
  11. Lines 328-330
  12. Materials and methods Section:

In 2.5 you must add more information on Receiver operating characteristic (ROC) since it is utterly important for your results and discussion. This will enhance the validity of the sensitivity and specificity, which is not clear.

  1. Why do you have a decrease in the control group in total sebum and anti-oxidant capacity (Figure 1)? Discuss it (Lines 301-303).

Does the fact that you have a decrease in uC mean that you don’t have the calculated decrease in uninvolved and psoriatic skin? Recalculate and explain the finding and statistics.

  1. Lines 307-310 are contradictory. Correct them.
  2. In accordance with questions 13 and 14: Does your research results finally agree with Darlenski or Pektas?
  3. Furthermore, Lines 310-311: “Furthermore, we propose a new, non-invasive tool to assess antioxidant capacity directly on the skin”.

Is this true according to the new explanation?

Author Response

In the manuscript by Trinidad Montero-Vilchez et al, on “Erythema increase predicts psoriasis improvement after phototherapy: an observational prospective study”, the authors have performed extensive research, but some methodological and outcome questions have risen.

Thank you for the comments

Also, probably due to the language barrier, many sentences and ideas are not clear or easy to understand. Additionally, some sentences are too long. The manuscript needs major revisions. Rephrase:

  1. Line 8

Line 8 in the manuscript is the email of one of the authors (salvadorarias@ugr.es (S.A.-S.). Did you mean another line?

  1. Lines 46-48

The sentence has been rephrased as recommended: Nevertheless, it is unknown the type of patient that would respond better to each treatment[13]. Moreover, tools to asses disease severity and treatment effectiveness are subjective[14].

  1. Line 60

It has been changed by As the development of psoriasis plaques results from the deregulation of epidermal keratinocytes and immunity cells

  1. Lines 69-71

It has been rephrased: To compare epidermal barrier function and skin homeostasis between normal skin (healthy participants), uninvolved psoriatic skin and psoriatic plaques

  1. Line 88 UVB-NB or NB-UVB Line 103?

NB-UVB. It has been corrected

  1. Lines 104-106.

It has been rephrased: Exposed subjects were evaluated before and after receiving the first phototherapy ses-sion and before and after the 15th phototherapy session. The starting dose for NB-UVB therapy and the dosage schedule were based on skin phototype following the actual guidelines[25]. The frequency was twice or three times weekly depending on the pa-tient's availability.

  1. Line 178

Differences in skin homeostasis parameters after one phototherapy session

Skin homeostasis parameter were modified after one phototherapy session

  1. Lines 270-272

It has been reprhased: ) A receiver operating characteristic (ROC) curve was created to determine the optimal cutoff value of erythema increases after one phototherapy session to predict PASI improvement in patients with psoriasis after fifteen phototherapy sessions

  1. Lines 279-280

It has been modified by It has been observed differences in skin homeostasis parameters between healthy skin, uninvolved psoriatic skin and psoriatic plaques

  1. Lines 291-293

It has been modified: The differences in TEWL and SCH values between psoriatic plaques and uninvolved psoriatic skin may be explained by a low AQP3 expression in plaques

  1. Lines 328-330

It has been rephased: To the best of our knowledge, it is unknown what parameters might predict clinical improvement in psoriatic patients treated with phototherapy

Moreover, an English native speaker, Charlotte Bower specialized in scientific translation, has review the English of the article.

  1. Materials and methods Section:

In 2.5 you must add more information on Receiver operating characteristic (ROC) since it is utterly important for your results and discussion. This will enhance the validity of the sensitivity and specificity, which is not clear.

Following your recommendations, we have included more information regarding ROC curves: To produce these ROC curves, the sensitivities and specificities for changes of erythema and SCH values after the first phototherapy that predict an improvement in PASI ≥ 3 after the fifteenth phototherapy session were tabulated and the graphical ROC curve was generated by plotting true positive rate (sensitivity) on the y-axis against false positive rate (1–specificity) on the x-axis for the various values tabulated. To select the optimal cut-off point, it was chosen the point nearest to the top-left most corner of the ROC curve, giving equal weight to the importance of sensitivity and specificity.

  1. Why do you have a decrease in the control group in total sebum and anti-oxidant capacity (Figure 1)? Discuss it (Lines 301-303).Does the fact that you have a decrease in uC mean that you don’t have the calculated decrease in uninvolved and psoriatic skin? Recalculate and explain the finding and statistics.

We have calculated changes in antioxidant capacity and sebum in healthy skin, uninvolved psoriatic skin and psoriatic plaque. Following your recommendations, we have tried to explain it in the discussion and this sentence has been added: This fact might mean that the time have also an impact in antioxidant capacity or that the sticks used might lose its capacity to measure the antioxidant capacity along the time.

  1. Lines 307-310 are contradictory. Correct them.

We have deleted the following sentence Moreover, high erythema increased are associated with phototherapy adverse events. We are not sure if it is your recommendation because line 307-310 in the pdf of the manuscript say Changes in SCH might be explained by the phototherapy effect on the inhibition of epidermal hyperproliferation. SCH and TEWL changes were greater at psoriatic plaques than at uninvolved psoriasis skin what might highlight a local effect on psoriasis plaques.

  1. In accordance with questions 13 and 14: Does your research results finally agree with Darlenski or Pektas?

Our results agree with Petkas research but it didn’t with Darlenskis ones as is commented in the text. Petkas assessed the total oxidant status and observed that it increased after phototherapy while we evaluated the antioxidant capacity and found that it decreased. In contrast Darlenski showed a decreased in a decreased in detoxifying activity of catalase, meaning a decreases in the oxidant status.

  1. Furthermore, Lines 310-311: “Furthermore, we propose a new, non-invasive tool to assess antioxidant capacity directly on the skin”. Is this true according to the new explanation?

Following your recommendation we have omitted this sentence

Reviewer 2 Report

Although, the presented paper is interesting, some changes are required.

  1. In the introduction, please add information on how advanced (light, moderate, severe) cases are phototherapy. Add national / international recommendations.
  2. In the abstract please change "thirty-eight patient" to number.
  3. In all comparisons in the both text of abstract and whole manuscript please add p value
  4. "In agreement with previous reports, this study shows that the whole epidermal barrier is affected in psoriatic patients, not just at psoriatic plaques [27]" please edit and write in 3rd os. singular.
  5. The title of Table S1 does not sound goods and should be rewritten.
  6. In Table S1, please provide all p values, also those statistically insignificant and mark significant ones, e.g. with an asterisk in the superscript.
  7. Please replace the entry "3.13 (3.13 SD)" with 3.13 ± 3.13 
  8. recommend shortening the title
  9. Table 1-3 please better explain in the text what the numerical values in front of and in parentheses mean. 10. Graphical abstract is suggested.

Author Response

Although, the presented paper is interesting, some changes are required.

Thank you for the comments.

  1. In the introduction, please add information on how advanced (light, moderate, severe) cases are phototherapy. Add national / international recommendations.

We have included information regarding international recommendations and the type of cases for which phototherapy is recommended. The following sentence has been added: Phototherapy is an effective, safe and low-cost therapy for mild-moderate psoriasis while many medical visits are needed to get psoriasis improvement[16]. It has been developed several types of light and lasers for treating psoriasis, being the narrowband ultraviolet light B (NB-UVB) the most frequently used. NB-UVB wavelengths ranges from 311to 313 nm. The starting dose is based on skin phototype or in minimal erythema dose (MED) and two or three sessions weekly are recommended

  1. In the abstract please change "thirty-eight patient" to number.

We have changed thirty-eight to number as recommended.

  1. In all comparisons in the both text of abstract and whole manuscript please add p value

The p value has been added in the abstract and whole manuscript.

  1. "In agreement with previous reports, this study shows that the whole epidermal barrier is affected in psoriatic patients, not just at psoriatic plaques [27]" please edit and write in 3rd os. singular.

We have rephrased the sentence as recommended: In agreement with previous reports, it has been found that the whole epidermal barrier is affected in psoriatic patients, not just at psoriatic plaques

  1. The title of Table S1 does not sound goods and should be rewritten.

The title of table S1 has been changed as recommended by Characteristics of the participants included in the study.

  1. In Table S1, please provide all p values, also those statistically insignificant and mark significant ones, e.g. with an asterisk in the superscript.

Following your recommendation all p values have been provided and the significant ones have been marked with an asterisk.

  1. Please replace the entry "3.13 (3.13 SD)" with 3.13 ± 3.13 

Following your recommendation all this entry and similar ones have been changed.

  1. I  recommend shortening the title

The title has been shorten as recommended and it has been changed by Erythema increase predicts psoriasis improvement after phototherapy

  1. Table 1-3 please better explain in the text what the numerical values in front of and in parentheses mean. 10. Graphical abstract is suggested

To explain what the numerical values in front of and in parentheses mean, we have added the following sentence Data are expressed as means (standard deviation). We have included a figure before and after 15 phototherapy sessions.

Round 2

Reviewer 1 Report

Thank you authors for revising the manuscript.

Hence, it is accepted in its current form

Reviewer 2 Report

The authors responded to my comments. Thank you and good luck !